# Comparison of Volatile Oil between the Fruits of *Amomum villosum* Lour. and *Amomum villosum* Lour. var. *xanthioides* T. L. Wu et Senjen Based on GC-MS and Chemometric Techniques

**DOI:** 10.3390/molecules24091663

**Published:** 2019-04-28

**Authors:** Hui Ao, Jing Wang, Lu Chen, Shengmao Li, Chunmei Dai

**Affiliations:** 1College of Pharmacy, Chengdu University of Traditional Chinese Medicine, Chengdu 611137, China; aohui2005@126.com (H.A.); wangjing2018@foxmail.com (J.W.); 2Innovative Institute of Chinese Medicine and Pharmacy, Chengdu University of Traditional Chinese Medicine, Chengdu 611137, China; 3School of Pharmacy, North Sichuan Medical College, Nanchong 637007, China; 4College of Basic medicine, Jinzhou Medical University, Jinzhou 121000, China

**Keywords:** *Amomum villosum* Lour., *Amomum villosum* Lour. var. *xanthioides* T. L. Wu et Senjen, gas chromatography–mass spectrometry (GC-MS), chemometric techniques, volatile oil, species distinction

## Abstract

Fructus Amomi (FA) is usually regarded as the dried ripe fruit of *Amomum villosum* Lour. (FAL) or *Amomum villosum* Lour. var. *xanthioides* T. L. Wu et Senjen (FALX.). However, FAL, which always has a much higher price because of its better quality, is often confused with FALX. in the market. As volatile oil is the main constituent of FA, a strategy combining gas chromatography–mass spectrometry (GC-MS) and chemometric approaches was applied to compare the chemical composition of FAL and FALX. The results showed that the oil yield of FAL was significantly higher than that of FALX. Total ion chromatography (TIC) showed that cis-nerolidol existed only in FALX. Bornyl acetate and camphor can be considered the most important volatile components in FAL and FALX., respectively. Moreover, hierarchical cluster analysis (HCA) and principal component analysis (PCA) successfully distinguished the chemical constituents of the volatile oils in FAL and FALX. Additionally, bornyl acetate, α-cadinol, linalool, β-myrcene, camphor, d-limonene, terpinolene and borneol were selected as the potential markers for discriminating FAL and FALX. by partial least squares discrimination analysis (PLS-DA). In conclusion, this present study has developed a scientific approach to separate FAL and FALX. based on volatile oils, by GC-MS combined with chemometric techniques.

## 1. Introduction

Fructus amomi (FA), also called Sharen, is a famous traditional Chinese medicine (TCM). It was firstly recorded as a medicinal resource in Yao Xing Lun (Tang Dynasty), and has a long medical history of more than one thousand years for treating gastrointestinal diseases and pregnancy-related diseases in China. In Chinese Pharmacopeia (2015 version), FA is defined as the dried ripe fruit of three ginger plants—*Amomum villosum* Lour., *A. villosum* Lour. var. *xanthioides* T. L. Wu et Senjen, and *A. longiligulare* T. L. Wu [1]—which has the effect of eliminating dampness and appetizing the stomach, warming the spleen and stopping diarrhea. The first two ginger plants are the main varieties in the Chinese market [2]. The dried ripe fruits of *A. villosum* Lour. (FAL) are called Yang Chun Sha, while those of *A. villosum* Lour. var. *xanthioides* T. L. Wu et Senjen (FALX.) are called Lv Ke Sha. Although the two species are used in the same way according to Pharmacopeia, it is generally acknowledged that FAL is more highly regarded by the general public, and considered one of the four most famous TCMs in the South of China, with a high price. In the Chinese market, FAL costs five to ten times more than FALX. Therefore, FALX. is often used as a counterfeit version of FAL because of their similar appearance and close genetic relationship. Precisely, they both originate from the genus *Amomum* in the Gingeraceae family, and possess indistinguishable shape, color, surface characteristics, odor and other characteristics. Therefore, it is imperative to clarify the differences between the two species and to obtain valuable additional evidence for identification.

To date, there are relatively few studies on the identification of FAL and FALX. Fourier Transform Infrared Spectroscopy (FTIR) was reported to differentiate FA (including FAL and FALX.) from other confusable varieties, but could not distinguish FAL from FALX. [3]. A previous study proved that the ITS-1 sequence could effectively identify FAL and FALX. [4], which indicated that the two herbs biologically differed from each other. However, because FA is the dried processed product of the fruit, most of the DNA is degraded and destroyed during the processing. It is difficult to extract high quality DNA, therefore, the DNA barcode is not suitable for the identification of dried FA.

FA is rich in volatile oil. Several studies have reported that the volatile oil of FA and its major compounds not only possessed anti-microbial, anti-inflammatory and analgesic activities, but also could be regarded as potential drugs for digestive diseases which cannot be effectively treated by chemicals, such as nonalcoholic fatty liver disease, 5-fluorouracil-induced intestinal mucositis and inflammatory bowel disease [5,6,7,8,9,10]. Therefore, volatile oil is regarded as the main active ingredient of FA. However, non-volatile compounds of FA showed few activities, according to the previous reports. Therefore, the volatile oil of FA is the quality control component in Chinese pharmacopoeia. Moreover, in order to differentiate FAL from FALX., it is necessary to find an efficient approach to distinguish the volatile oil of FAL and FALX. by their chemical composition. Steam distillation combined with gas chromatography-mass spectrometry (GC-MS) is used as the routine method for the analysis of the volatile oils of FA [11,12,13]. However, until now no study could show the chemical difference between the volatile oils of FAL and FALX.

Conventional mutual chemical comparison cannot find elements which result in quality variance. Chemometrics, based on computer and modern computing technology, is a new interdisciplinary subject. In recent years, chemical analysis combined with chemometric methods such as hierarchical cluster analysis (HCA), principal component analysis (PCA) and partial least squares discrimination analysis (PLS-DA) have been widely used in the identification, qualitative character, quality control and efficacy relationship of herbs [14,15,16,17,18,19], which seems to be an ideal tool for this problem.

Therefore, in this study, a comprehensive strategy combining GC-MS analysis and chemometric methods was firstly proposed to compare FAL with FALX. Precisely, GC-MS combined with chemometric methods including HCA and PCA was employed for the identification of the volatile oils in the two confused species, while unpaired student’s–T-test (T-test) and PLS-DA were utilized to discover the potential chemical markers for discriminating these two herbal medicines. The aim of the present study is to investigate the chemical differences in the volatile oils between the two confused species of FA, by GC-MS in general and in detail.

## 2. Results and Discussion

### 2.1. Fingerprints of FAL and FALX.

According to total iron chromatography (TIC), the chemical composition of the two species was similar in general but still had some difference. The number of peaks of FALX. from 17 min to 25 min was higher than that of FAL. The peak, whose retention time (RT) was 21.345 min) identified as cis-nerolidol, appeared only in the chromatogram of FALX., but its relative percentages were too low, at only 0.23% to 0.48%. The chromatograms of FAL possessed 16 common peaks while those of FALX. had 17 common peaks. Among the above peaks, 13 common peaks, whose total areas accounted for above 90% of the volatile constituents and represented the chemical characteristic of the samples well, could be found in the chromatograms of FAL and FALX. The results are shown in Table 1 and Figure 1 and Figure 2.

In order to find the marker compounds for chemical comparison of FAL and FALX., an unpaired T-test was employed for analyzing the relative contents of the common peaks of FA. The *p*-value was set as the filtering standard in order to maintain the contents. The relative contents of the two compounds, camphor and bornyl acetate, regarded as the main chemical components in the essential oils of FA, were significantly different in FAL and FALX. In the FAL samples, the relative contents of bornyl acetate were the highest, ranging from 41.32% to 60.20%, with an average of 49.16 ± 5.13%, while that of camphor was the second highest, ranging from 16.84% to 28.90% with an average of 22.81 ± 3.79%, and the relative contents of both were above 70% with an average of 70.97 ± 2.14%. Meanwhile, in FALX. samples excluding S16, the relative contents of camphor were the highest ranging from 29.07% to 44.76% with an average of 36.13 ± 4.18%, and those of bornyl acetate were the second highest, ranging from 15.85% to 31.63% with an average of 24.51 ± 4.10%. The sum of the relative contents of the two was above 56% with an average of 60.64 ± 2.54%. In general, the relative contents of bornyl acetate in FAL were significantly higher than in FALX., but those of camphor in FAL were significantly lower than in FALX. In addition, the relative contents of other common peaks in the two species of FA were also obviously different. The relative contents of β-myrcene, α-phellandrene, d-limonene, terpinolene, linalool, camphor, α-terpineol, α-cadinol and α-santalol in FAL were 2.49~3.38%, 0.29~0.39%, 6.70~8.31%, 0.23~0.28%, 0.25~0.85%, 1.89~4.66%, 0.26~0.36%, 0.16~0.37% and 0.16~0.34%, successively. The relative contents of those in FALX. were 3.89~6.07%, 0.21~0.30%, 8.10~11.35%, 0.18~0.23%, 1.76~3.62%, 3.79~8.09%, 0.31~0.57%, 0.80~1.79% and 0.40~0.87%, respectively. Among these figures, the relative contents of α-phellandrene and terpinolene were significantly higher in FAL than in FALX., while β-myrcene, d-limonene, linalool, camphor, borneol, α-terpineol, α-cadinol and α-santalol were significantly lower in FAL than in FALX. Furthermore, whether in FAL or in FALX., bornyl acetate and camphor were the two highest percentage compounds and accounted for more than half of all oil samples. Bornyl acetate, the quality control component of FA in Chinese pharmacopeia, always accounted for a higher percentage in FAL than in FALX. Wei et al [20] found that bornyl acetate accounted for more than camphor in FAL according to nine reports from 1985–2000. Comparatively, the percentage of camphor was significantly higher in FALX. than in FAL (*p* < 0.05). Moreover, the ratio of the percentages of bornyl acetate to camphor was significantly higher in FLA than in FLAX. Therefore, it was concluded that FAL was a plant characterized by bornyl acetate while FALX. seemed to be a plant whose oil was mainly dependent on camphor. As was already known, bornyl acetate exhibited anti-inflammatory [21,22,23], analgesic, anti-tumor [24], whitening, anti-oxidative [25] and immune-regulatory [26] effects while camphor showed various pharmacological effects including anti-tussive [27], anti-oxidative [28], anti-fungal [29], anti-wrinkle [30] and wound-healing [31] activities. However, no report regarding the pharmacological differences of FAL and FALX. has been seen, which hinders the development and rational utilization of FA. Further study is needed to compare the bioactivities of the two species.

The results are shown in Figure 3.

### 2.2. HCA

HCA is a clustering technique that measures either the difference or the similarity between the objects to be clustered. Based on the relative contents of each component in the essential oil, the samples of FA with close similarities will be crudely classified into the same cluster by HCA. All the test samples were performed using a Ward method to visualize the differences and/or similarities among samples through Euclidean distance. The results showed that the samples were clustered into two groups. The first group included S1~S9 (FAL) and the second group covered S10~S20 (FALX.). In other words, FAL and FALX. could be distinguished based on the composition of the oils. The results are shown in Figure 4.

### 2.3. PCA

To provide more information about differentiation of the origins of FA samples, PCA was performed based on the 13 common peak areas. It was applied in order to reduce the number of variables (13 variables corresponding to the components in essential oil from FAL and FALX.) to a smaller number of new derived variables (PCs) that adequately summarize the original information. The first two PCs explained approximately 83.9% of the original data variability (Figure 5). The score scatter plot is displayed in Figure 6. From the PCA scatter plot, FAL (S1~S9) and FALX. (S10~S20) were divided into two areas respectively, which was similar to the result of cluster analysis. Moreover, the dots which presented S1~S9 were relatively nearer to each other, suggesting a closer relationship among the nine batches of FAL. The dots of S10~S20 were relatively scattered, indicating diversification of the 11 batches of samples. It was indicated that the chemical composition of FALX. was less stable compared with FAL. This was probably because the FALX. samples were collected from many parts of Asia, for instance, Vietnam, Thailand and Myanmar and Yunnan, compared with FAL, which was cultivated only in the south and southeast of China. The results are shown in Figure 5.

### 2.4. PLS-DA

Both HCA and PCA could clearly clarify the composition of the two species, but failed to find out the variables for sample classification. Therefore, a supervised PLS-DA technique was used to visualize the variations among these samples. R2X, R2Y and Q2Y levels obtained by PLS-DA were 0.840, 0.972 and 0.962 respectively, which were suitable for fitness and prediction. Based on the PLS-DA, a loadings plot was drawn to exhibit the contribution of each variable to the discrimination of FAL and FALX. As shown in Figure 6, the 13 common peaks were listed in order according to their contribution value. Eight components with variable important plot (VIP) values greater than or equal to 1.00 were selected as the potential markers, namely bornyl acetate, α-cadinol, linalool, β-myrcene, camphor, d-limonene, terpinolene and borneol. Notably, the unpaired T-test proved that all of the selected constituents could distinguish FAL from FALX. Therefore, these markers could be used for species identification and quality control of FA. Considering that the volatile oil of FA exhibited various activities, it is worth studying whether the above compounds are responsible for these activities. The results are shown in Figure 6.

In conclusion, GC-MS analysis combined with chemometric technologies can distinguish FAL from FALX. As is known, species is the key internal factor affecting the quality of Chinese herbs, and the correct species is the primary premise to ensure their security and effectiveness. Attention should be paid to the chemical differences among different species, and their similarities and differences in biological activity and clinical efficacy need to be further revealed.

## 3. Methods

### 3.1. Plant Materials

A total of 20 batches of FA including 9 batches of FAL (batch number 2016001FAL, 2016002FAL, 2016003FAL, 2016004FAL, 2016005FAL, 2016006FAL, 2016007FAL, 2016008FAL, and 20160009FAL) and 11 batches of FALX. (batch number 2016001FALX., 2016002FALX., 2016003FALX., 2016004FALX., 2016005FALX., 2016006FALX., 2016007FALX., 2016008FALX., 20160009FALX., 20160010FALX. and 20160011FALX.) were collected from Chengdu Lotus Pond Chinese Herbal Medicine Market. FAL were cultivated in Yangchun and Gaozhou in Guangdong Province, Baise in Guangxi, and Ruili, Menghai and Mengla in Yunnan. The origins of FALX. were Vietnam, Thailand and Myanmar in addition to Yunnan. Twenty batches of samples, authenticated by Lu Chen (Associate Professor of Chengdu University of Traditional Chinese Medicine) were deposited in the chemical laboratory.

### 3.2. Solvents and Chemicals

Analytical grade *n*-hexane was purchased from Beijing Chemical works (Beijing, China). Anhydrous sodium sulfate was provided by Chemical Reagent Co. Ltd. of SINOPHARM (Shanghai, China). *n*-alkane (C_8_–C_30_) series were provided by Sigma Co. Ltd (Shanghai, China).

### 3.3. Steam Distillation for Volatile Oil

About 30 g FAL or 50 g FALX. was put into a 1000 mL distillation flask. Ten times that amount of water was added and volatile oil distillation apparatus was set according to Chinese Pharmacopoeia. The oil was distilled for 6 h, obtained from the condenser, and dried over anhydrous sodium sulfate. The oil yields were calculated in milliliters of oil per 100 g of FA. Twenty micrograms of the obtained essential oil was introduced into 1.5 mL autosampler vials, then was filtered through a 0.22 µm filter and the final volume of the extract was adjusted to 1.0 mL with *n*-hexane.

### 3.4. GC-MS Analysis

GC-MS analysis was performed on an Agilent Technologies apparatus 7890A-5975C with HP-5 MS capillary column (30 m × 0.25 mm, 0.25 μm film thickness). Helium was applied as the carrier gas at a constant flow rate of 1 mL/min. The injector temperature was 250 °C and interface temperature was 280 °C. The initial oven temperature was kept at 60 °C., then it was gradually raised to 124 °C at 4 °C/min, to 196 °C at 8 °C/min, to 260 °C at 10 °C /min and finally kept for 2 min. The spectrometer operated at 70 ev with the full scan style. The injection mode was split with a 60:1 (*v*/*v*) ratio. Retention indexes (RI) were calculated for the common components using a homologous series of *n*-alkanes injected in conditions identical to those of the samples. Identification of the common components was based on their RI relative to *n*-alkanes and comparison with the published literature. Then, the components of the volatile oil were positively identified using National Institute of Standards and Technology (NIST) 14.0 Mass Spectra Database. The semi-quantitative analysis of volatile compounds was performed by comparing their peak areas in the GC-MS total ion chromatogram. The percentage compositions of compounds were calculated by area normalization method.

### 3.5. Data Analysis

Statistical analysis was carried out using an unpaired T-test by GraphPad Prism 7 (GraphPad Software Inc., La Jolla, CA, USA)). Moreover, these data were also analyzed and processed by HCA, PCA and PLS-DA using SPSS13.0 (SPSS Inc., Chicago, IL) or SIMCA P11.0 (Umetrics, Umea, Sweden). Results were expressed in means ± SEM, and the level of *p* < 0.05 was considered as statistically significant.

## 4. Conclusions

In the present study, an efficient strategy for species identification of FA was developed by GC-MS analysis and chemometric methods. GC-MS fingerprints showed that FAL and FALX. were similar and had 13 common peaks. Yet at the same time, the difference was also very significant. First of all, the main component of the volatile oils was different. Bornyl acetate had the largest relative peak area in FA ranging from 41.32% to 60.20%, while in FALX. camphor accounted for the highest percentages ranging from 29.07% to 44.76%. Moreover, some markers with important identification values were found. Cis-nerolidol was only found in FAL, not in FALX, and the other eight components, bornyl acetate, α-cadinol, linalool, camphor, β-myrcene, d-limonene, terpinolene and borneol, could also be used as the distinguishing components by the unpaired T-test. Moreover, chemometric analysis based on the GC-MS spectrum including HCA, PCA and PLS-DA showed that there were significant differences in the volatile components of FAL and FALX., and the samples from the same variety were clustered together. In short, GC-MS analysis combined with chemometric methods could efficiently distinguish FAL and FALX.

Conclusively, the authors believe that chemical composition lays the foundation for the material basis of efficacy. Among Chinese herbs, some species look similar in extrinsic features and are used in the same way, but they perhaps have different ingredients and effects to some extent. GC-MS analysis combined with chemometric technologies can provide an accurate method to distinguish these similar Chinese herbs containing volatile oils.

## Figures and Tables

**Figure 1 molecules-24-01663-f001:**
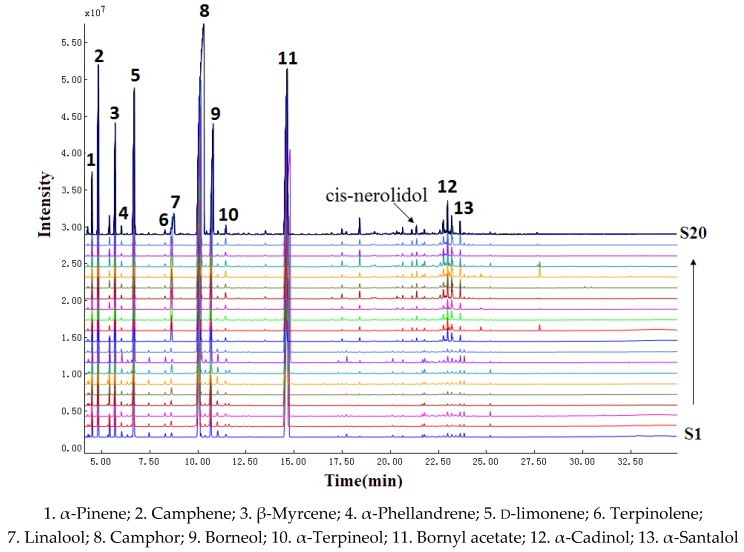
Gas chromatography–mass spectrometry (GC-MS) chromatogram of 20 batches of FA from two species.

**Figure 2 molecules-24-01663-f002:**
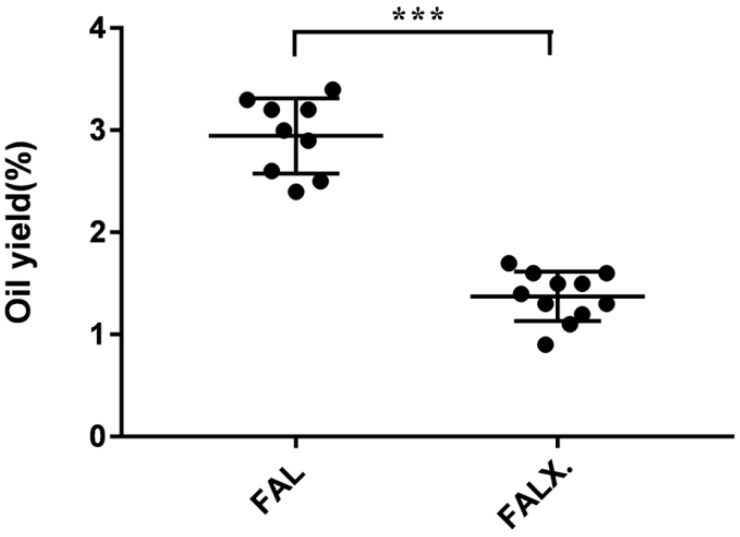
Oil yields of 20 batches of FA from two species. Data were presented as mean ± SEM. *** *p* < 0.001, compared with *Amomum villosum* Lour. var. *xanthioides* T. L. Wu et Senjen (FALX.).

**Figure 3 molecules-24-01663-f003:**
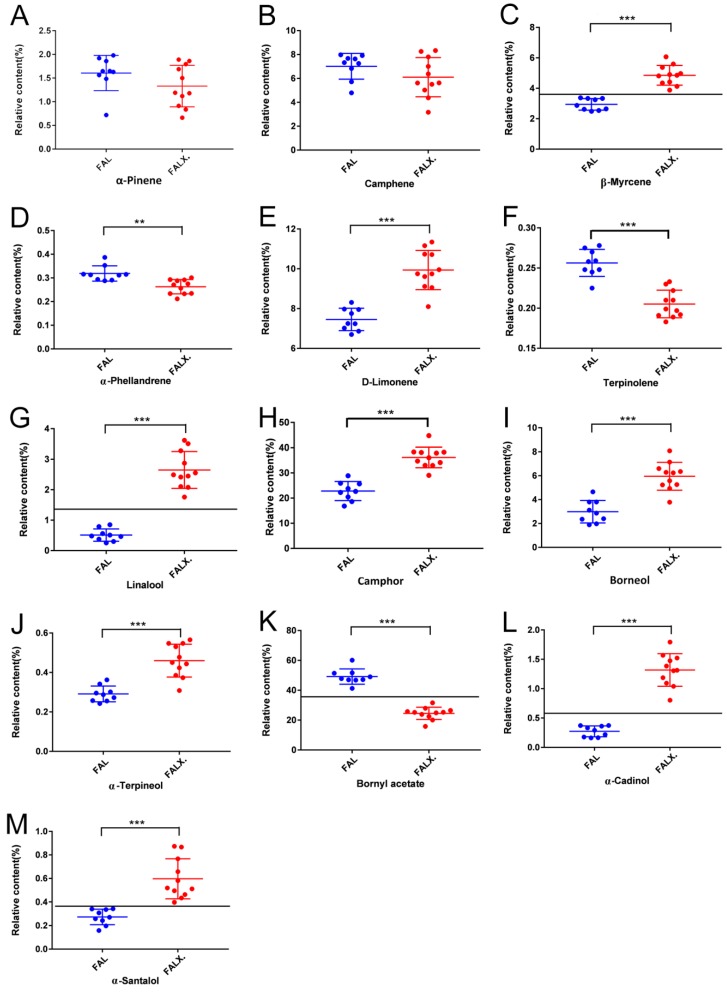
Relative contents of the common peaks in total ion chromatography (TIC) of FA from two species (**A**) α-pinene, (**B**) Camphene, (**C**) β-Myrcene, (**D**) α-Phellandrene, (**E**) d-Limonene, (**F**) Terpinolene, (**G**) Linalool, (**H**) Camphor, (**I**) Borneol, (**J**) α-Terpineol, (**K**) Bornyl acetate, (**L**) α-Cadinol and (**M**) α-Santalol. Data were presented as mean ± SEM. *** *p* < 0.001, ** *p* < 0.01compared with FALX. Blue dots meant the relative content of the common peaks in each FAL sample; while red dots meant those in each FALX. sample.

**Figure 4 molecules-24-01663-f004:**
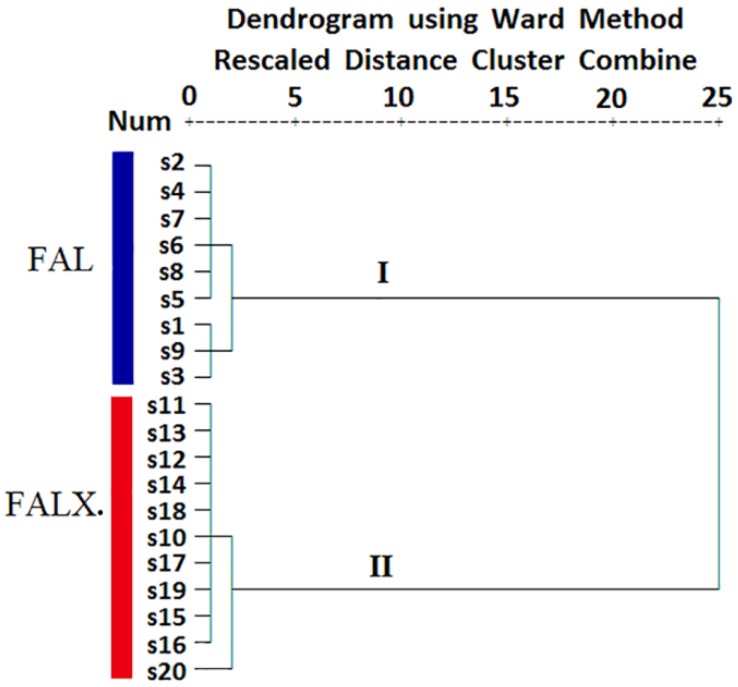
Hierarchical cluster analysis (HCA) of 20 FA samples from two species using Ward’s method based on the Euclidean distance.

**Figure 5 molecules-24-01663-f005:**
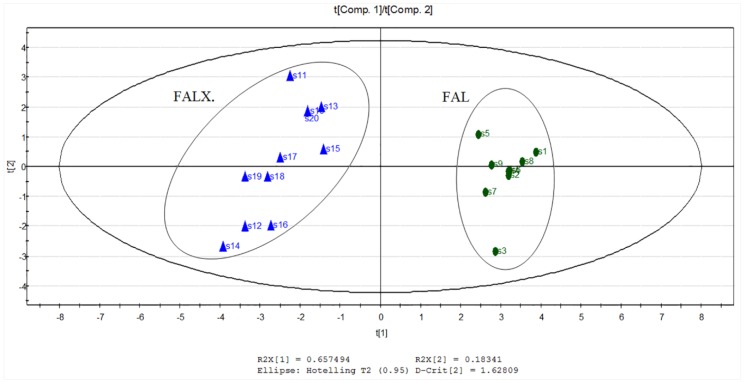
Score plot of principal component analysis of 20 samples of FA from two species.

**Figure 6 molecules-24-01663-f006:**
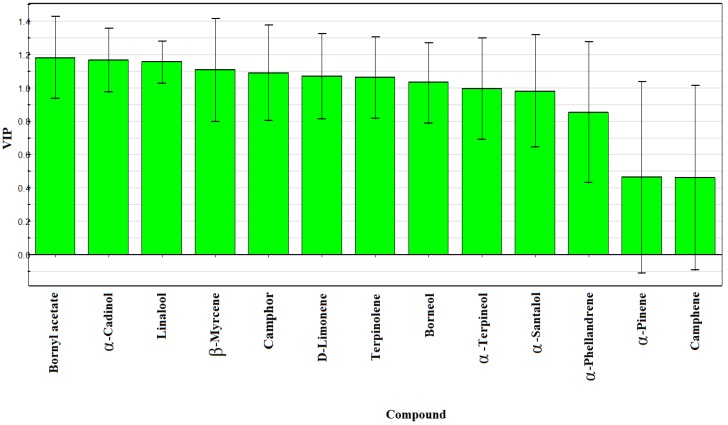
Variable important plot (VIP) of identified compounds of 20 samples of FA from two species based on partial least squares discrimination analysis (PLS-DA).

**Table 1 molecules-24-01663-t001:** The relative contents of the common peaks of Fructus amomi (FA) from the two species.

No.	Retention Time	Molecular Weight	Component	Molecular Formula	RI	RI′	S1	S2	S3	S4	S5	S6	S7	S8	S9	S10	S11	S12	S13	S14	S15	S16	S17	S18	S19	S20
1	4.487	136.2	α-Pinene	C_10_H_16_	917	931	1.92	1.65	0.72	1.63	1.98	1.86	1.65	1.56	1.49	1.69	1.86	0.84	1.89	0.66	1.5	0.91	1.18	1.12	1.19	1.79
2	4.810	136.2	Camphene	C_10_H_16_	935	946	7.92	6.84	4.80	7.33	7.98	7.64	5.72	7.25	7.67	8.34	8.26	4.38	7.80	3.17	6.38	5.03	5.63	5.51	5.62	7.01
3	5.681	136.2	β-Myrcene	C_10_H_16_	990	988	3.25	2.54	2.65	2.49	3.34	2.88	2.62	3.38	3.33	4.97	6.07	4.13	5.37	4.42	4.90	3.89	5.6	4.85	4.82	4.35
4	6.022	136.2	α-Phellandrene	C_10_H_16_	1006	1003	0.35	0.31	0.29	0.29	0.31	0.29	0.31	0.39	0.32	0.29	0.29	0.21	0.30	0.23	0.29	0.23	0.28	0.23	0.27	0.26
5	6.663	136.2	d-limonene	C_10_H_16_	1030	1033	7.96	6.87	7.02	6.70	7.98	7.24	7.25	7.76	8.31	10.72	11.35	9.12	11.17	9.75	9.96	9.06	9.61	9.75	10.74	8.10
6	7.439	136.2	Terpinolene	C_10_H_16_	1075	1078	0.28	0.26	0.25	0.25	0.24	0.26	0.27	0.28	0.23	0.21	0.19	0.21	0.23	0.23	0.22	0.21	0.19	0.19	0.21	0.18
7	8.610	154.1	Linalool	C_10_H_18_O	1103	1102	0.38	0.56	0.31	0.51	0.79	0.48	0.85	0.25	0.46	2.49	2.11	3.62	2.87	3.51	2.45	2.42	2.08	2.55	3.28	1.76
8	10.069	152.2	Camphor	C_10_H_16_O	1148	1141	18.65	25.68	16.84	25.95	28.90	22.29	23.82	22.7	20.46	32.97	36.08	38.28	38.35	37.88	34.16	29.07	33.02	38.14	34.73	44.76
9	10.651	154.2	Borneol	C_10_H_18_O	1169	1168	1.98	3.80	3.12	3.80	2.87	1.89	4.66	2.37	2.41	6.29	7.15	5.24	3.79	5.58	4.94	6.25	6.35	5.27	6.61	8.09
10	11.433	154.2	α-Terpineol	C_10_H_18_O	1194	1189	0.29	0.26	0.26	0.24	0.32	0.36	0.34	0.30	0.27	0.44	0.31	0.55	0.48	0.57	0.39	0.53	0.42	0.45	0.55	0.37
11	14.692	196	Bornyl acetate	C_10_H_20_O_2_	1288	1287	51.68	47.15	60.20	46.83	41.32	49.07	47.02	47.81	51.46	25.31	20.11	26.52	22.49	24.87	27.95	31.64	25.75	25.21	23.89	15.85
12	22.956	222.2	α-Cadinol	C_15_H_26_O	1666	1652	0.29	0.18	0.37	0.16	0.33	0.16	0.22	0.36	0.37	1.19	1.57	1.39	0.80	1.52	1.10	1.31	1.79	1.48	1.32	1.04
13	23.609	220.2	α-Santalol	C_15_H_24_O	1707	1705	0.31	0.21	0.34	0.16	0.24	0.27	0.26	0.34	0.34	0.52	0.43	0.66	0.49	0.87	0.58	0.87	0.46	0.51	0.77	0.41
			Total percentages of common compounds				95.26	96.31	97.17	96.34	96.6	94.69	94.99	94.75	97.12	95.43	95.78	95.15	96.03	93.26	94.82	91.42	92.36	95.26	94	93.97
			V (mL)				0.96	1.02	0.87	0.78	0.99	0.75	0.9	0.96	0.72	0.85	0.75	0.55	0.45	0.6	0.7	0.75	0.8	0.65	0.65	0.8
			Oil yield (%)				3.2	3.4	2.9	2.6	3.3	2.5	3.0	3.2	2.4	1.7	1.5	1.1	0.9	1.2	1.4	1.5	1.6	1.3	1.3	1.6
			Others				4.74	3.69	2.83	3.66	3.4	5.31	5.01	5.25	2.88	4.57	4.22	4.85	3.97	6.74	5.18	8.58	7.64	4.74	6	6.03
			Number of total peaks				30	27	26	27	24	27	31	36	21	27	27	25	25	31	31	40	38	25	33	42
			Number of identified peaks				28	27	25	26	24	27	29	34	20	27	26	25	24	29	27	38	35	25	32	38

RI, retention index; RI′, retention index from the literature.

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
