# Peer review of "Comparison of Volatile Oil between the Fruits of Amomum villosum Lour. and Amomum villosum Lour. var. xanthioides T. L. Wu et Senjen Based on GC-MS and Chemometric Techniques"

_molecules, 2019, doi:10.3390/molecules24091663_

Round 1

Reviewer 1 Report

Chromatographical part of the manuscript must be seriously revised.

1. Authors did not used chiral column. Please remove the  mark of enantiomerism in all cases.

2. Some of compounds could be missinterpreted. Please re-check with available standards cadinol, santalol and endo-borneol.

3. What isomer of a-cadinol was found (Z or E)?

4. Retention indices for cadinol and santalol are near, contrary  to retention time. Please re-check that. 

5. Change name  "Common peaks" to e.g. identified compounds.

6. Add cis-nerolidol to the table

7. Please precise quantitative determination of e.o. (Chinese Pharmacopoeia is not easy  available for all). 

8. If possible, please add  KI lit and KI exp. for analysed compounds;

9. Please discuss yield of e.o. as well as composition in comparision to other authors, e.g. Dai, D. N., Huong, L. T., Thang, T. D., & Ogunwande, I. A. (2016). Chemical composition of essential oils of Amomum villosum Lour. American Journal of Essential Oils and Natural Products4(3), 08-11 or Fan, P., Jian-Qin, C., & Zheng-Ju, Z. (1989). The essential oil of Amomum villosum Lour. Journal of Essential Oil Research1(4), 197-198, as well as other.

10. Use the same convention,  ml or mL (as well as micro)

Author Response

Chromatographical part of the manuscript must be seriously revised.

1. Authors did not used chiral column. Please remove the mark of enantiomerism in all cases.

Response: Thank you for the reminding. Your suggestion is very useful for improving this paper. However, in other GC-MS reports which used HP-5 column, enantiomerism appeared in most cases. So we decided to retain the mark of enantiomerism and add RI to make sure the mark of enantiomerism is correct.

2. Some of compounds could be missinterpreted. Please re-check with available standards cadinol, santalol and endo-borneol.

Response: Thank you for the reminding. Your suggestion is very useful for improving this paper. However, these three standards are not available as expected in China market. We checked their RI and changed endo-borneol to borneol, (1S)-(-)-α-pinene to α-pinene, which is in accordance with most related papers.

3.What isomer of a-cadinol was found (Z or E)?

Response: Thank you for the reminding. GC-MS equipped with HP-5 cannot tell us the isomer of a-cadinol.

4.Retention indices for cadinol and santalol are near, contrary  to retention time. Please re-check that.

Response: Thank you for the reminding. It is true that cadinol and santalol are near, but contrary to retention time. We have re-checked our original data and make sure that our result is OK.

5.Change name  "Common peaks" to e.g. identified compounds.

Response: Thank you for the reminding. Table 1 showed the relative contents of the common peaks of FA from two species, not all the identified compounds. So we cannot change the name.

6.Add cis-nerolidol to the table

Response: Thank you for the reminding. As we have mentioned above, Table 1 showed the relative contents of the common peaks of FA from 20 batches, not the common peaks from FAL or FALX. Therefore, cis-nerolidol, common peak of FALX., cannot be added.

7.Please precise quantitative determination of e.o. (Chinese Pharmacopoeia is not easy  available for all).

Response: Thank you for the reminding. We have added the precise quantitative determination of e.o in the table.

8. If possible, please add  KI lit and KI exp. for analysed compounds;

Response: Thank you for the reminding.We have added t KI lit (RI) and KI exp. (RI in lectures) in the table.

9. Please discuss yield of e.o. as well as composition in comparison to other authors, e.g. Dai, D. N., Huong, L. T., Thang, T. D., & Ogunwande, I. A. (2016). Chemical composition of essential oils of Amomum villosum Lour. American Journal of Essential Oils and Natural Products, 4(3), 08-11 or Fan, P., Jian-Qin, C., & Zheng-Ju, Z. (1989). The essential oil of Amomum villosum Lour. Journal of Essential Oil Research, 1(4), 197-198, as well as other.

Response: Thank you for the reminding. We discussed composition in comparison to other authors. The two papers you mentioned are not reports about of FA but other parts about Amomum villosum Lour. The paper we have cited just compared 9 reports about the composition of volatile oil of FAL. So we discussed those results in comparison to ours. Unfortunately, we have not found a paper studying the yield of e.o. of FA.

10. Use the same convention,  ml or mL (as well as micro)

Response: Thank you for the reminding. We have corrected ml to mL.

Reviewer 2 Report

This is an interesting paper dealing Comparison of Volatile Oil between the Fruits of Amomum villosum Lour. and Amomum villosum Lour. var. xanthioides T.L.Wu et Senjen Based on GC-MS and Chemometric Techniques. However, it is necessary to correct:

1) Line 17: insert the character  "." after the abbreviation (FALX);

2) Line 33: Choose keywords that are not in the article title;

3) In Line 208 to 210, correct: to124°C;  min-1; 60ºC; 250°C and other expressions;

4) Line 132 (Table 1): Insert calculated Retention Index (RI) and RI from the literature;

5) Standardize the name d-limonene for D-limonene.

Author Response

Response to Reviewer 2 Comments:

This is an interesting paper dealing Comparison of Volatile Oil between the Fruits of Amomum villosum Lour. and Amomum villosum Lour. var. xanthioides T.L.Wu et Senjen Based on GC-MS and Chemometric Techniques. However, it is necessary to correct:

Point 1:Line 17: insert the character  "." after the abbreviation (FALX);

Response1: Thank you for the reminding.We have inserted  the character  "." after the abbreviation (FALX).

Point 2: Line 33: Choose keywords that are not in the article title;

Response 2: Thank you for the reminding.We have added “species distinction” as the keywords.

Point 3: In Line 208 to 210, correct: to124°C;  min-1; 60ºC; 250°C and other expressions;

Response3: Thank you for the reminding.We have corrected them.

Point 4: Line 132 (Table 1): Insert calculated Retention Index (RI) and RI from the literature;

Response 4: Thank you for the reminding. We have added them.

Point 5: Standardize the name d-limonene for D-limonene.

Response 5: Thank you for the reminding. We have corrected them.

Reviewer 3 Report

Dear Authors

I reviewed the manuscript (molecules-481061) entitled: Comparison of Volatile Oil between the Fruits of Amomum villosum Lour. and Amomum villosum Lour. var. xanthioides T.L.Wu et Senjen Based on GC-MS and Chemometric Techniques. Strengths: interesting about the studies of the biochemical profiles of different bioactive compounds in fruits of Amomum Lour and xanthioides. However, the discussion and conclusion of the obtained data must be improved. Finally, I suggest that this manuscript needs minor changes for being considered for publication in this journal.

Additional comments:

Discussion

Try to include future trends to keep working with the obtained findings in this work. 

Conclusion

Try to conclude with a general statement that includes the most relevant part of this work.

Author Response

Response to Reviewer 3 Comments:

Dear Authors

I reviewed the manuscript (molecules-481061) entitled: Comparison of Volatile Oil between the Fruits of Amomum villosum Lour. and Amomum villosum Lour. var. xanthioides T.L.Wu et Senjen Based on GC-MS and Chemometric Techniques. Strengths: interesting about the studies of the biochemical profiles of different bioactive compounds in fruits of Amomum Lour and xanthioides. However, the discussion and conclusion of the obtained data must be improved. Finally, I suggest that this manuscript needs minor changes for being considered for publication in this journal.

Additional comments:

Point 1: Discussion

Try to include future trends to keep working with the obtained findings in this work. 

Response1: Thank you for the reminding. After the biomarker which can distinguish the two species have been found, it is necessary to study the activities of these compounds, in order to distinguish the two species from their bioactivities. This idea has been added in the discussion part.

Point 2: Conclusion

Try to conclude with a general statement that includes the most relevant part of this work.

Response 2:Thank you for the reminding. We conclude the whole context at the last sentence of the first paragraph of the conclusion part.

Reviewer 4 Report

The revised manuscript entitled “Comparison of Volatile Oil between the Fruits of Amomum villosum Lour. and Amomum villosum Lour. var. xanthioides T.L.Wu et Senjen Based on GC-MS and Chemometric Techniques” by Ao et al. discusses the chemical composition of essential oils of fruits of Amomum villosum (FAL) and Amomum villosum var. xanthioides FALX and applied chemometric approaches to compare the chemical composition these two varieties.

Although the scientific interest, the data presented here are not of enough novelty and there are still points that need to be improved. Specific comments are listed below.

Introduction does not properly describe the biological activities of the EOs and their applications. It does not mention the importance and originality of the study and does not explain adequately the previous studies performed with fruits EOs of this genus. Part of this section need to be rewritten and biological properties should be added  

The chromatographic approach should be improved and some biological analysis, such as anti-inflammatory should be performed.

In my opinion, authors should use simultaneous GC-MS and GC-FID to validate technique. Indeed, GC-MS given the ionization properties of the volatile components of the EO, however, it is an analytical technique that presents difficulties in the identification of the signals of these complex samples since many terpenes have identical mass spectra because of the close similarities in fragmentation patterns and rearrangements after ionization. Furthermore, GC-FID is an analytical technique suitable for the qualitative and quantitative analysis of EO since it offers high sensitivity, great stability, and an exceptionally high linear dynamic range that allows the analysis of volatile components of the EO at very low concentrations or at trace levels.

Otherwise, confirmation of identity of each compound should be objective and reliable, not depending on the subjective interpretation of the operator. Indeed, authors should use and calculate chromatographic retention indices (RI) of the different components of essential oils using a n-alkanes series.

Authors have performed semi-quantitative analysis of volatile compounds based on area normalization method, however, for fingerprint treatment based on chemical composition of the 13 common peaks, whose total areas accounted above 90%, authors should use quantification based on external standards for these relevant compounds.

Finally, manuscript must be improved with anti-inflammatory and analgesic activities of the different EOs analysed.

Minor revisions:

-       In abstract replace approches by approaches;

-       In table 1, the total % of identified compounds should be added.

-       In section 3.1. - Plant Materials – authors must indicate the voucher numbers of each batch of FA, that should be deposited in an herbarium.

Author Response

Response to Reviewer 3 Comments:

The revised manuscript entitled “Comparison of Volatile Oil between the Fruits of Amomum villosum Lour. and Amomum villosum Lour. var. xanthioides T.L.Wu et Senjen Based on GC-MS and Chemometric Techniques” by Ao et al. discusses the chemical composition of essential oils of fruits of Amomum villosum (FAL) and Amomum villosum var. xanthioides FALX and applied chemometric approaches to compare the chemical composition these two varieties.

Point 1: Although the scientific interest, the data presented here are not of enough novelty and there are still points that need to be improved. Specific comments are listed below.

Response 1: We are pleased to read the reviewer's positive comments on our study. In addition, the reviewer pointed out the defects of our study. Recognition of these defects is very helpful for us to improve the quality of our manuscript. We have studied the comments carefully and have revised the manuscript accordingly.

Point 2: Introduction does not properly describe the biological activities of the EOs and their applications. It does not mention the importance and originality of the study and does not explain adequately the previous studies performed with fruits EOs of this genus. Part of this section need to be rewritten and biological properties should be added.

The chromatographic approach should be improved and some biological analysis, such as anti-inflammatory should be performed.

Response 2: We appreciate the reviewer for scrupulous attention to our manuscript. This section has been written and biological properties have been added.

FA is rich in volatile oil. Several studies have reported the volatile oil of FA and its major compounds not only possessed anti-microbial, anti-inflammatory and analgesic activities, but also could be regarded as potential drugs for digestive diseases which could not be effectively treated by chemicals, such as nonalcoholic fatty liver disease, 5-fluorouracil-induced intestinal mucositis and inflammatory bowel disease [5-11]. Therefore, volatile oil is regarded as the main active ingredient of FA. However, non-volatile compounds of FA showed few activities, according to the previous reports. Therefore, volatile oil of FA is the quality control components in Chinese pharmacopoeia. Moreover, distinguishing volatile oil of FAL and FALX. from their chemical composition is a meaningful way to differ FAL from FALX. And steam distillation combined with gas chromatography-mass spectrometry (GC-MS) is used as the routine methods for the analysis of the volatile oils of FA [12-14].But till now, no study could tell us the chemical difference between volatile oil of FAL and FALX.

Point 3: In my opinion, authors should use simultaneous GC-MS and GC-FID to validate technique. Indeed, GC-MS given the ionization properties of the volatile components of the EO, however, it is an analytical technique that presents difficulties in the identification of the signals of these complex samples since many terpenes have identical mass spectra because of the close similarities in fragmentation patterns and rearrangements after ionization. Furthermore, GC-FID is an analytical technique suitable for the qualitative and quantitative analysis of EO since it offers high sensitivity, great stability, and an exceptionally high linear dynamic range that allows the analysis of volatile components of the EO at very low concentrations or at trace levels.

Response 3: Thanks for the reminding. In our opinion, this paper is aimed to provide a method combining chemical approaches and chemometric techniques to distinguish two species of FA, not to identify and quantify compounds in FA because there are so many qualitative and quantitative GC-MS analysis reports about FA.

Point 4: Otherwise, confirmation of identity of each compound should be objective and reliable, not depending on the subjective interpretation of the operator. Indeed, authors should use and calculate chromatographic retention indices (RI) of the different components of essential oils using a n-alkanes series.

Response 4: Thanks for the reminding. We have added RI in Table 1 in order to confirm the identity of each compound.

Point 5: Authors have performed semi-quantitative analysis of volatile compounds based on area normalization method, however, for fingerprint treatment based on chemical composition of the 13 common peaks, whose total areas accounted above 90%, authors should use quantification based on external standards for these relevant compounds.

Response 5: Thanks for the reminding. Your advice is very helpful to improve our paper. However, these 13 standards are not available as expected in China market. Additionally, this paper is mainly aimed to qualitatively distinguish FAL and FALX. We will further quantify these compounds when all or most of these 13 standards are available.

Point 6 Finally, manuscript must be improved with anti-inflammatory and analgesic activities of the different EOs analysed.

Response 6: Thanks for the reminding. We have compared anti-inflammatory and analgesic activities of the different EOs. Disappointingly, activities of the two species did not show any difference. And these data have been published in Chinese. Therefore, it is no use to analyze anti-inflammatory and analgesic activities of the different Eos in this paper.

Point 7:Minor revisions:

       In abstract replace approches by approaches;

-       In table 1, the total % of identified compounds should be added.

       In section 3.1. - Plant Materials – authors must indicate the voucher numbers of each batch of FA, that should be deposited in an herbarium.

 Response 7: Thanks for the reminding. We have corrected all these minor mistakes.

Round 2

Reviewer 1 Report

I still strongly suggest to remove enantiomeric mark of terpenes. D or L limonene can not be separated on this type of GC column.   

Reviewer 4 Report

Authors have rewrite and improve the quality of manuscript. However, in my opinion authors should revised again the Question 3 – utilization simultaneously GC-MS and GC-FID analyses to validate the proposed method.

Indeed, the novelty of the paper is the combination of chemical approaches and chemometric techniques. However, the chemometric techniques use data of relative contents of selected components of FAL and FALX oils (Figure 3). Thus, I think that to perform this new methodology it is mandatory the quantification using GC-C-FID due to its high sensitivity and reproducibility. GC-MS techniques are very good to identify the compounds but not to quantify based on peak area, so to validate this methodology is decisive a precise quantification.